# MOC Composites for Construction: Improvement in Water Resistance by Addition of Nanodopants and Polyphenol

**DOI:** 10.3390/polym15214300

**Published:** 2023-11-01

**Authors:** Anna-Marie Lauermannová, Ondřej Jankovský, Adéla Jiříčková, David Sedmidubský, Martina Záleská, Adam Pivák, Milena Pavlíková, Zbyšek Pavlík

**Affiliations:** 1Department of Inorganic Chemistry, Faculty of Chemical Technology, University of Chemistry and Technology, Technická 5, 166 28 Prague, Czech Republic; anna-marie.lauermannova@vscht.cz (A.-M.L.); ondrej.jankovsky@vscht.cz (O.J.); adela.jirickova@vscht.cz (A.J.); david.sedmidubsky@vscht.cz (D.S.); martina.zaleska@fsv.cvut.cz (M.Z.); 2Department of Materials Engineering and Chemistry, Faculty of Civil Engineering, Czech Technical University in Prague, Thákurova 7, 166 29 Prague, Czech Republic; adam.pivak@fsv.cvut.cz (A.P.); milena.pavlikova@fsv.cvut.cz (M.P.)

**Keywords:** MOC, tannic acid, nanoadditives, mechanical strength, softening coefficient

## Abstract

The topic of modification of magnesium oxychloride cement (MOC) using specific functional additives is very much pronounced in the research of alternative building materials. This study deals with the co-doping of MOC by 1D and 2D carbon nanomaterials in order to improve its mechanical properties while using tannic acid (TA) as a surfactant. Furthermore, the effect of TA on MOC also improves its water resistance. As a filler, three size fractions of standard quartz sand are used. The proposed types of MOC-based composites show promising results considering their mechanical, macro- and microstructural, chemical, and hygric properties. The use of 1D and 2D nanoadditives and their mixture enables the improvement in the flexural strength and particularly the softening coefficient, which is the durability parameter characterizing the resistance of the prepared materials to water. After immersion in water for 24 h, the compressive strength of all tested specimens of modified composites was higher than that of the reference composite. Quantitatively, the developed co-doped composites show mechanical parameters comparable to or even better than those of commonly used Portland cement-based materials while maintaining high environmental efficiency. This indicates their potential use as an environmentally friendly alternative to Portland cement-based products.

## 1. Introduction

In the attempts to mitigate the carbon footprint originating from the construction and building industry, researchers turn to alternative composite materials that could possibly replace the commonly used Portland cement (PC). Among such materials, magnesium oxychloride cement (MOC) and composites on its base have played a major role in past decades. The environmental advantage of MOC compared to PC is based on its ability to capture CO_2_ while forming carbonates [1]. It also manifests up to a 75% lower impact on resources [2]. Generally, MOC is formed in four main phases, of which Phase 5 is the most researched, mostly due to its simple synthesis, rapid hardening, and excellent mechanical properties [3,4,5].

Recent developments in the research on MOC mainly cover two approaches. The first one is the utilization of waste fillers and modifiers, and it is based on the ability of MOC to contain a large amount of aggregate while maintaining good mechanical performance. This approach further broadens the environmental sustainability of MOC-based materials, as it usually concerns wastes, which would otherwise need to be landfilled. Among the most used additives in this regard are various types of ash, waste wood, gypsum, sludge, and many others [6,7,8,9,10,11]. The second approach is based on the functional improvement in the properties of the composites based on MOC. For this purpose, the MOC matrix is modified using specific types of materials and additives. The most pronounced material properties in this regard are durability, water resistance, and mechanical performance. The studies concerning this approach usually result in highly functional materials based on MOC, which are applicable in variable conditions. Davras et al. studied the effect of mixing ratios during the production of MOC-based lightweight composites using hydrogen peroxide, potassium iodide, and methylcellulose [12]. The proposed ratio of the raw materials enabled the production of an ultra-lightweight MOC composite. Another type of lightweight, high-performance, MOC-based composite was proposed by Abd-El-Raoof et al. [13]. The basis of this experiment was the modification of the MOC matrix using polymers. Yan et al. studied the influence of multiple inorganic salts on hydration reactions occurring during MOC formation. This study demonstrated that such salts can promote the reaction rate, resulting in a higher amount present in MOC Phase 5 and a shortened curing time [14]. The studies focusing on the improvement in water resistance of MOC usually concern the application of functional additives, such as silica fume/nano-silica, hydroxyacetic acid, soluble phosphates, tannin, or carbon-based nanomaterials [15,16,17,18,19].

Recently, the introduction of carbon-based nanomaterials into construction composites has been a highly researched topic. The main contribution of such additives is their high potential to improve the physical and mechanical properties of the designed composite materials, especially in hydration behavior, strength, durability, and electrical and thermal conductivity. Furthermore, specific carbon-based nanomaterials can also provide specific functional properties and smart functions. All of these enhancements can be attributed to the unique properties and microstructure of these types of nanomaterials [20]. The influence of carbon-based nanomaterials on the hydration process is based on the facilitation of the pozzolanic reaction. This phenomenon is caused by the formation of nucleation sites where the cement hydrate phases sediment and grow [21,22,23]. The improvement in mechanical properties is based on the use of carbon-based nanomaterials as nano-reinforcement. The enhancement rate is relatively high, especially considering the generally low proportions of the nanomaterial used. The compressive, tensile, flexural, and fracture strengths, as well as the elastic moduli are among the properties improved [24,25]. The increase in durability is based on the refined pore structure and the reduced pore connectivity, which hinders the transport of aggressive ions and harmful agents. Furthermore, carbon-based nanomaterial-enhanced composites have been proven to have increased durability to chloride and sulfate ions, carbon dioxide molecules, and decalcification agents [26,27,28,29]. The improvement in thermal and electrical conductivity is based on the excellent thermal and electrical properties of the carbon-based nanomaterials themselves, which translate into the properties of the composite into which they are introduced [30,31,32]. The smart functions provided by the cement-based composites are mainly represented in self-sensing cement or environmental monitors for the detection of moisture or chloride content [33,34,35,36].

The enhancement of the composite material can also be further improved by using a combination or hybrid of multiple types of carbon-based nanomaterials. Li et al. studied the influence of the combination of graphene oxide sheets and single-walled carbon nanotubes. This research has shown the significance of the co-effect of these two types of carbon-based nanomaterials compared to their individual use. Furthermore, their influence on the particle size of the matrix crystals was assessed, as was their influence on the mechanical parameters [37]. Lu et al. studied the dispersion of such a combination of nanoadditives showing that graphene oxide can work as a surfactant for the homogenization of carbon nanotubes in a water suspension. This effect is based on increased electrostatic repulsion. As a result, more load is transferred from the matrix to the nano-reinforcement, resulting in enhanced compressive and flexural strength [38]. The combination of multi-walled carbon nanotubes and graphene sheets was proven to be useful in the enhancement of fracture toughness and microhardness of the prepared composites. These effects were assigned to a decrease in the porosity of the designed composite and a possible bridging effect of the nanoadditives, which resulted in the mitigation of crack propagation [39]. Yoo et al. have also shown the positive influence of combined carbon-based nanomaterials on the electrical properties of construction composites. This research has shown how the degradation of electrical properties can be decelerated using multi-walled carbon nanotubes, graphene nanoplatelets, and nanofibers [40].

The most pronounced challenge in the introduction of carbon-based nanomaterials into construction composites is their homogenous dispersion in the cementitious matrix. It was previously shown that these additives can show their full potential only when their agglomeration is omitted [41,42], which is caused by the Van der Waals forces present in their atomic structure [43,44,45,46]. The formation of agglomerates or clumps can result in weak spots in the microstructure of the composite, causing increased susceptibility to defects [47,48]. To ensure proper homogenization of the carbon-based nanomaterials, both specific incorporation methods and agents are used. Among such methods, ultrasonication is the most commonly applied [49]. As dispersing agents, mostly polycarboxylate-based surfactants such as methylcellulose or anionic superplasticizers are used. Their effect is based on steric hindrance or static charge repulsion [50,51,52]. As an environmentally friendly surfactant, tannic acid was previously studied [53]. The basis of its function as a surfactant lies in its adsorption onto the carbon-based nanomaterial creating a π–π interaction between the aromatic rings of tannic acid and carbon atoms in the structure of the nanomaterial. This process increases the steric repulsion between the individual carbon nanomaterial particles, which makes them easier to disperse [54,55,56]. The introduction of tannic acid into construction composites was recently studied by Fang et al. [57]. The experiments conducted showed that tannic acid can reduce the total porosity of the mortar, especially concerning the capillary pores with a size between 10 and 50 nm. This results in improved mechanical performance. Furthermore, tannic acid was proven to improve the cohesion in the hardened cement-based product due to its increased ability to cross-link the hydration phases via various types of interactions, such as hydrogen and ionic bonding or hydrophobic interactions [58,59]. Moreover, as tannic acid is a naturally occurring polyphenolic compound, it can be considered somewhat eco-friendly and possibly renewable [60].

This study concerns the research of MOC-based composites enhanced by the combination of various types of carbon-based nanomaterials while being simultaneously doped with tannic acid. The use of tannic acid in construction composites based on PC was previously studied; however, this type of modifier was not introduced into the eco-friendly MOC-based matrix. The expected effect of the tannic acid on MOC-based composites doped with multiple types of carbon-based nanomaterials is dual: First, it is presumed to work as a functional additive for the improvement in the water resistance of the designed composites due to its positive effect on the microstructure, and especially the porosity and density of the MOC-based matrix. Second, it is applied as a surfactant for homogenous dispersion of the applied carbon-based nanoadditives. The proposed composites should therefore manifest high values of mechanical parameters, which are further enhanced by the carbon-based nanoadditives, as well as the high water resistance ensured by the tannic acid. Such composites can be useful in construction, especially in specifically shaped or prefabricated elements.

## 2. Materials and Methods

For the preparation of the MOC-based composite samples, the following chemicals were used: MgCl_2_∙6H_2_O (>99%, Lachner s.r.o., Neratovice, Czech Republic); MgO (>98%, Penta s.r.o., Prague, Czech Republic); graphene nanoplatelets with a declared specific surface area of 750 m^2^·g^−1^ and a particle size of <2 µm (2D Sigma Aldrich, St. Louis, MO, USA); multi-walled carbon nanotubes TNIM8 with a declared purity of >95%, a specific surface area of >60 m^2^·g^−1^, and a length of <10 µm (1D, TimesNano, Chengdu, China); tannic acid (TA, Carl Roth GmbH + Co. KG, Karlsruhe, Germany); and quartz sand with three size fractions of 0–0.5, 0.5–1.0, and 1.0–2.0 mm, named PG1, PG2, and PG3, respectively (Filtrační písky, s.r.o., Chlum, Czech Republic). The designation of the samples was set as follows: REF for reference MOC sample with quartz sand filler; TA for MOC sample with quartz sand filler and tannic acid; 1D-TA for MOC sample with quartz sand filler, multi-walled carbon nanotubes, and tannic acid; 2D-TA for MOC sample with quartz sand filler, graphene nanoplatelets, and tannic acid; and 1D-2D-TA for MOC sample with quartz sand filler, multi-walled carbon nanotubes, graphene nanoplatelets, and tannic acid. Furthermore, a sample named PASTE consisting of only MOC with no filler was prepared. The fresh mixture properties of the specific samples are summarized in Table 1. The synthesis of the samples consisted of the preparation of an aqueous solution of MgCl_2_∙6H_2_O, in which the functional additives such as tannic acid, graphene nanoplatelets, and/or multi-walled carbon nanotubes were dispersed using a mechanical rotor-stator homogenizer Ultra-Turrax T-18 (IKA, Königswinter, Germany) for 5 min at a speed of 10,000 rpm. For the samples REF and PASTE, this step was not implemented. After that, the MgO powder, the homogenized MgCl_2_ solution/suspension of additives in MgCl_2_ solution, and the quartz sand filler were mixed together in a planetary type of mortar mixer. First, a low-speed regime (140 rpm) was used for 1 min, and then a high-speed regime (285 rpm) was used for 1 min. After that, the mixer bowl and paddle were scraped, and the mixing continued for 3 min at the high-speed regime. The prepared mixtures were then poured into prismatic molds with dimensions of 40 × 40 × 160 mm^3^, where they were left for 24 h. After that, they were demolded and cured for 27 days at *T* = (23 ± 2) °C and *RH* = (50 ± 5)%.

The samples were cured for 28 days and then used for multiple types of analyses. Their phase and chemical composition, microstructure and morphology, micro- and macrostructural properties, mechanical parameters, and hygric properties were studied. The whole experimental process is depicted in a scheme in Figure 1 together with photographs displaying the selected tests performed. The details of the experimental techniques used are also presented in the Appendix A. More experimental details can be found in our previous publications aimed at the research of MOC-based materials [61,62].

## 3. Results

In this study, five types of MOC-based composites were prepared. The influence of adding 1D and 2D carbon-based nanomaterials individually and in combination, as well as tannic acid was studied. A photograph of the prepared samples is shown in Figure 2.

Prior to testing the composite samples, the phase composition of the PASTE sample prepared without any filler or additive was determined via XRD. The diffraction pattern (see Figure 3) shows the presence of two phases. The majority of the sample consists of MOC Phase 5 (ICDD 04-014-8836) [63] with the main reflection at 2θ = 11.896°. The minor crystalline phase present in the pattern is chlorartinite (ICDD 04-015-1149) with the main reflection at 2θ = 7.576°, which is a resultant phase of the CO_2_ capture reaction occurring on the surface of MOC [64].

The XRD analysis of the prepared MOC-based composites revealed the presence of two major crystalline phases in all of the samples. First, silicon oxide in the form of quartz (ICDD 04-016-2085) with the main reflection at 2θ = 26.649° was present [65]. This phase can be attributed to the presence of quartz sand used as a filler in all of the composite samples. Second, the presence of MOC Phase 5 was confirmed, similarly to the paste sample. For the samples with tannic acid and carbon-based nanomaterials, the additives were not determined in the diffraction patterns, mainly due to their very low content. The XRD patterns of all of the prepared MOC-based composite samples are shown in Figure 4.

The chemical composition of the prepared MOC-based composites was studied using EDS. This method provided both the elemental maps (see Figure 5), as well as the content of each present element in wt.% (see Table 2). Present in all of the samples were carbon, oxygen, magnesium, silicon, and chlorine. The quantity of each element corresponded sufficiently with the intended composition of the composite; however, it should be noted that the content obtained from EDS can vary as the analysis considers only a very small area of the fracture surface of the analyzed sample. The element patterns show the distribution of the quartz sand particles (maps of Si) and the MOC-based matrix (maps of Mg and Cl). As oxygen is present in both quartz sand and MOC, its pattern is not determinative concerning these two phases.

SEM was used to study the microstructure of the prepared samples. The typical microstructure of MOC Phase 5 was very well visible on the micrographs of the PASTE sample (see Figure 6). The MOC Phase 5 needles with a length between 1 and 5 μm and a width of ~0.5 μm were very well visible at the highest magnification.

The micrographs of the microstructure of the prepared MOC-based composites are shown in Figure 7. Especially at a higher magnification, it can be seen that the presence of MOC Phase 5 in the form of needle-shaped crystals was proved. The micrographs also show how the use of tannic acid and carbon-based nanomaterials influence the shape of the MOC needles. For all of the samples containing tannic acid, thinning of the needles is apparent. This thinning was caused by the formation of a gel phase on the surface of the needles due to a surface reaction between tannic acid and MOC. Furthermore, the sample containing both types of carbon-based nanomaterials and tannic acid shows slight bending or even curling in the MOC needles.

The macro- and microstructural parameters of the tested composites are introduced in Table 3. The data were obtained from samples cured for 28 days and represent the mean value from the measurement of five samples. The changes in bulk density, specific density, and porosity were minimal, and taking into account the expanded combined uncertainty of the test methods used, they can be considered insignificant or even negligible from a practical point of view. The minimal variation in the measured fundamental structural parameters shows evidence of the positive effect of tannic acid used as a surfactant in avoiding the agglomeration of nanoadditive particles, and thus their homogeneous distribution in the MOC matrix. Moreover, TA itself did not negatively affect the formation of the MOC structure. The assessment of the microstructural parameters by mercury intrusion porosimetry (MIP) indicated that the porous structure was refined by the co-doping of the composite by the combination of 1D and 2D nanoparticles, which was also noticed based on the obtained macrostructural parameters. This was also observed in the composites 1D-TA and 2D-TA. The determined average pore size is a characteristic of medium capillaries, which affect mechanical strength and water permeability. The cumulative pore size distribution is shown in Figure 8. The shape of the pore size curves correspond well with the calculated microstructural parameters. Quantitatively, the measured porosity values are low, as demonstrated by both test techniques, i.e., the combination of helium pycnometry with the gravimetric determination of bulk density and MIP. In general, as capillary porosity affects mechanical strength and stiffness and reduces water ingress, the results obtained for nano-modified composites with TA can be considered promising.

The mechanical parameters of the investigated composites are summarized in Figure 9. The composites exhibited high mechanical strength and stiffness. The flexural/compressive strength ratio was about 30%, which is much higher than that reported for PC concrete, whose flexural strength makes up approximately 10–20% of its compressive strength [66]. In general, the addition of tannic acid as well as nanoadditives slightly decreased the compressive strength and slightly increased the flexural strength of the prepared composites. The reduction in compressive strength was lesser for composites with TA and 2D nanoadditives compared to that for TA- or TA +1D-modified materials. The increase in the flexural strength is the result of three effects: (i) the high intrinsic tensile strength of the used nanoadditives themselves [67], (ii) the bridging effect of the 2D nanodopant, and (iii) the decrease in the volume of pores having a diameter of >0.1 μm. The last effect was not identified for the 2D-TA sample, but was very visible for the materials TA, 1D-TA, and 1D-2D-TA. Dynamic Young’s modulus was almost identical for all samples.

Water absorption values are presented in Figure 10. A slight decrease in water absorption was obtained for all enhanced composites; however, the major positive influence of the dopants was proved after water immersion for 24 h. Figure 9 shows that after water immersion, the highest compressive strength was obtained for the co-doped sample 1D-2D-TA, reaching 56.9 MPa, which is much higher than the 43.7 MPa obtained for the reference sample. The dramatic drop in the compressive strength of the reference material is the result of the decomposition of MOC Phase 5 into Mg(OH)_2_, which has much lower mechanical strength [68]. Within the water-induced structural changes, the mechanical interlocking of a stable needle-like crystallized Phase 5 is damaged, which is the reason for the loss of mechanical integrity. This mechanism is generally attributed to the pure water resistance of MOC-based products [69,70]. The softening coefficient was 51.4% for REF, 77.3% for TA, 69.8% for 1D-TA, 68.1% for 2D-TA, and 73.8% for 1D-2D-TA, confirming the significant influence of tannic acid addition on the compressive strength of all the composites. The increase in the softening coefficient is very promising for the practical use of the prepared materials. The improved durability is a result of the solidified structure of the composites, the effect of chelation between TA and Mg^2+^ ions, and their reinforcement against crack propagation induced by chemical changes in the precipitated MOC Phase 5. Since TA has a high molecular weight and is rich in phenolic hydroxyl groups, it forms a chelate with Mg^2+^ that forms a gel-like shell on the surface of precipitated products of MOC hydration [71]. This shell prevents contact of the MOC Phase 5 needles with water, thus improving water resistance. The effective reinforcement of the MOC matrix is attributed to the fact that the used nanoadditives can bridge the pores between sand particles and MOC Phase 5 crystals. Since the pores have an average pore size of about 0.05 μm, both the graphene particles and the multi-walled carbon nanotubes allow the reinforcement of these weak points in the composite microstructure. Moreover, in the case of water damage, they can partially withstand the generated tensile stress, thus at least partially eliminating the failure rate and loss of integrity.

## 4. Conclusions

Tannic acid was used in MOC-based composites in combination with 1D and 2D carbon-based nanoadditives to improve the technical parameters and durability of composites made of MOC and quartz sand with a major focus on the water resistance. The two nanomaterials used, graphene nanoplatelets and multi-walled carbon nanotubes, and their combination positively influenced the water resistance in the hardened materials, as well as the flexural strength when incorporated into composite blends. All of the modified MOC-based composites manifested an increase in the softening coefficient, which is a crucial indicator of their durability against water-induced damage. The softening coefficient was improved by tannin and carbon nanodopants by 35.8%, 32.5%, and 43.6% in MOC composites with graphene, multi-walled carbon nanotubes, and their combination, respectively. The highest increase in the softening coefficient was obtained for the samples TA (77.3%) and 1D-2D-TA (73.8%). These values are more than sufficient for the application of the designed composites in the field where PC-based composites are used. Regarding flexural strength, the improvements were 9.3%, 6.9%, and 0.8% for 1D-TA, 2D-TA, and 1D-2D-TA samples, respectively.

The application of TA enabled the avoidance of the agglomeration of nanoparticles, and thus their homogeneous distribution in the MOC–sand matrix, which is a crucial and problematic step in the introduction of nanoadditives in construction materials. Since MOC is generally considered to be more environmentally friendly due to its ability to sequester CO_2_ from the environment of the application, the developed composites effectively modified by nanoadditives can be a potential alternative to the Portland cement-based products that dominate the current construction market. In the prospective application of the researched composites, it is necessary to consider their excellent mechanical strength, high compressive/flexural strength ratio, and good resistance to excessive moisture.

## Figures and Tables

**Figure 1 polymers-15-04300-f001:**
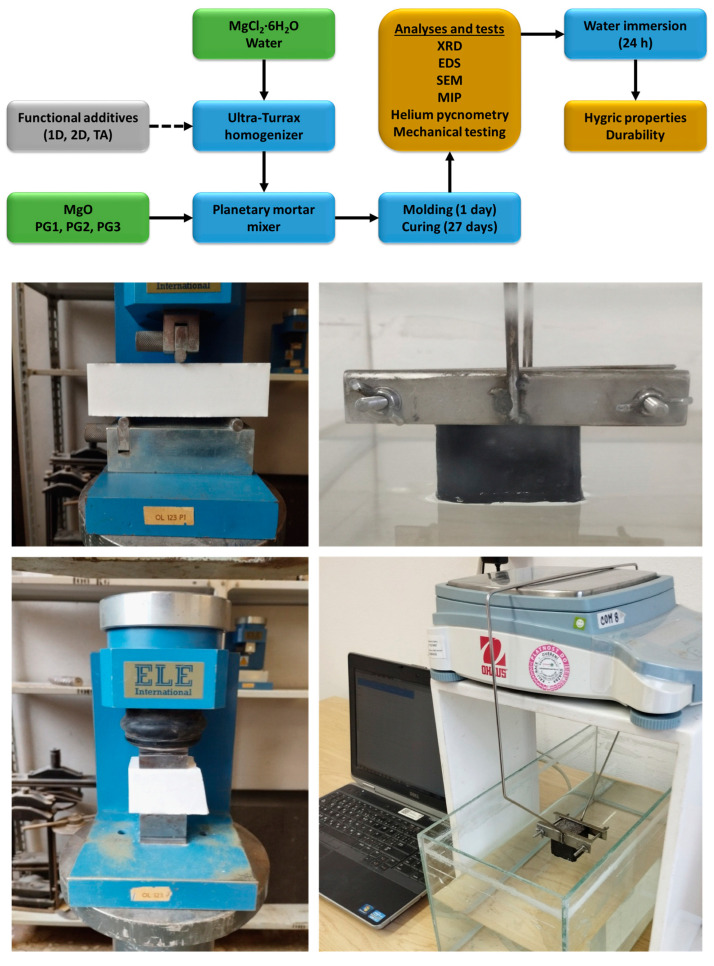
Scheme of the experimental process of MOC-based composite preparation and characterization (**up**) and photographs depicting the selected test (**down**)—flexural and compressive strength (**left**) and water absorption (**right**).

**Figure 2 polymers-15-04300-f002:**
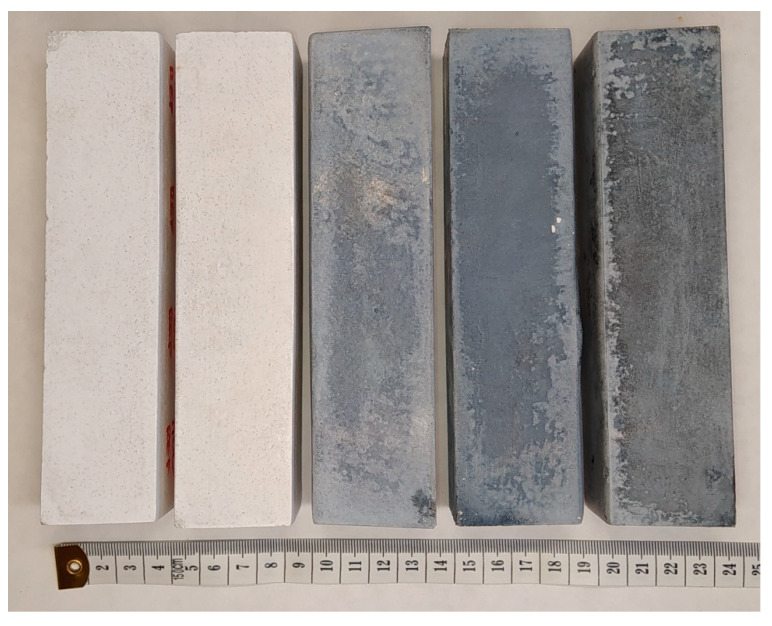
Photograph of prepared MOC-based composite samples (from left: REF, TA, 1D-TA, 2D-TA, and 1D-2D-TA).

**Figure 3 polymers-15-04300-f003:**
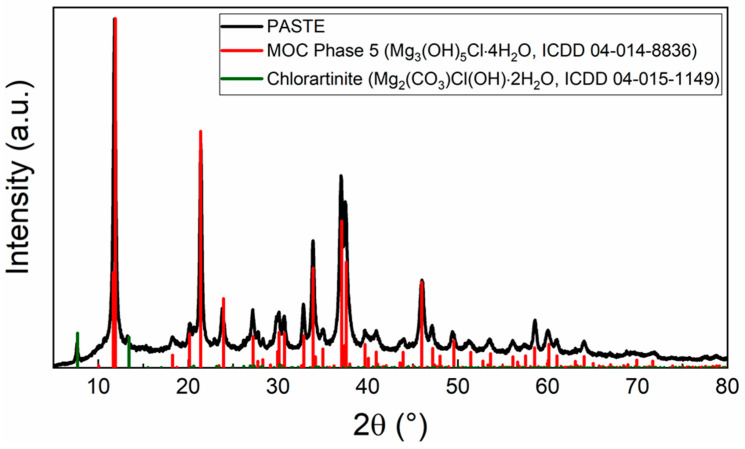
XRD pattern of the prepared MOC paste.

**Figure 4 polymers-15-04300-f004:**
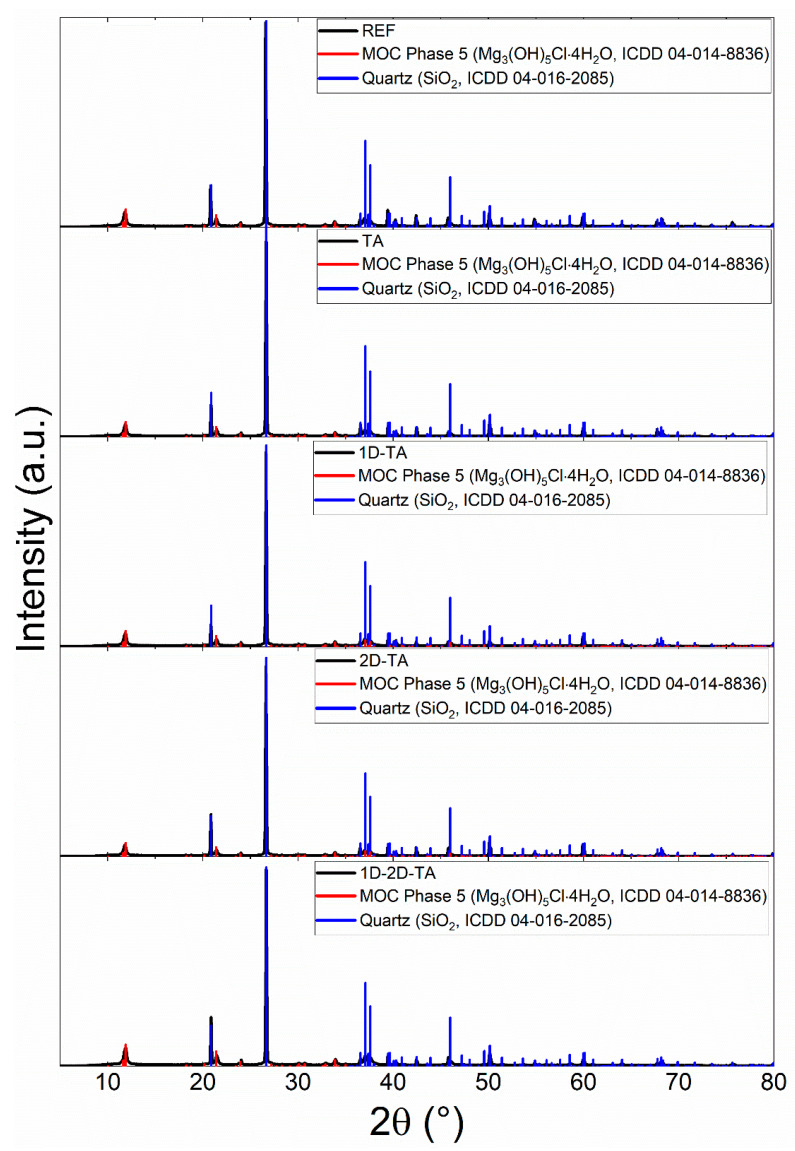
XRD patterns of prepared MOC-based composite samples.

**Figure 5 polymers-15-04300-f005:**
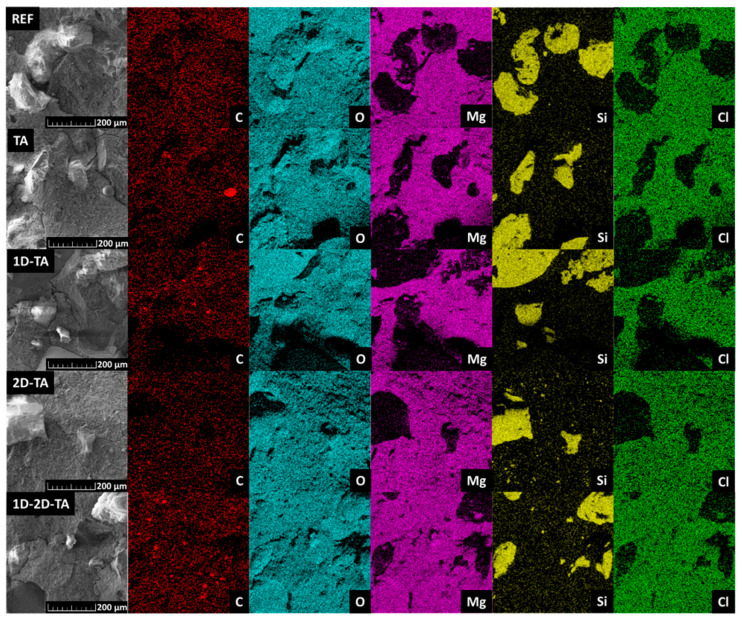
Elemental maps of prepared MOC-based composites obtained from EDS.

**Figure 6 polymers-15-04300-f006:**
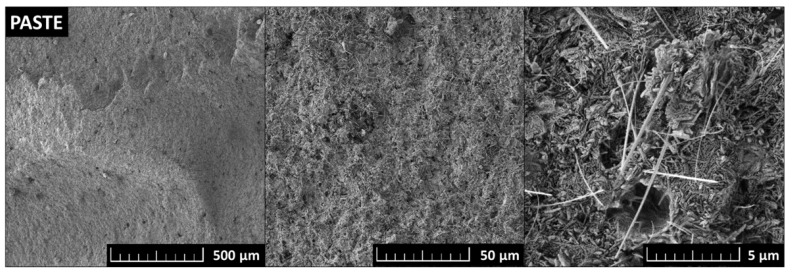
SEM micrographs of typical MOC Phase 5 microstructure studied on PASTE sample.

**Figure 7 polymers-15-04300-f007:**
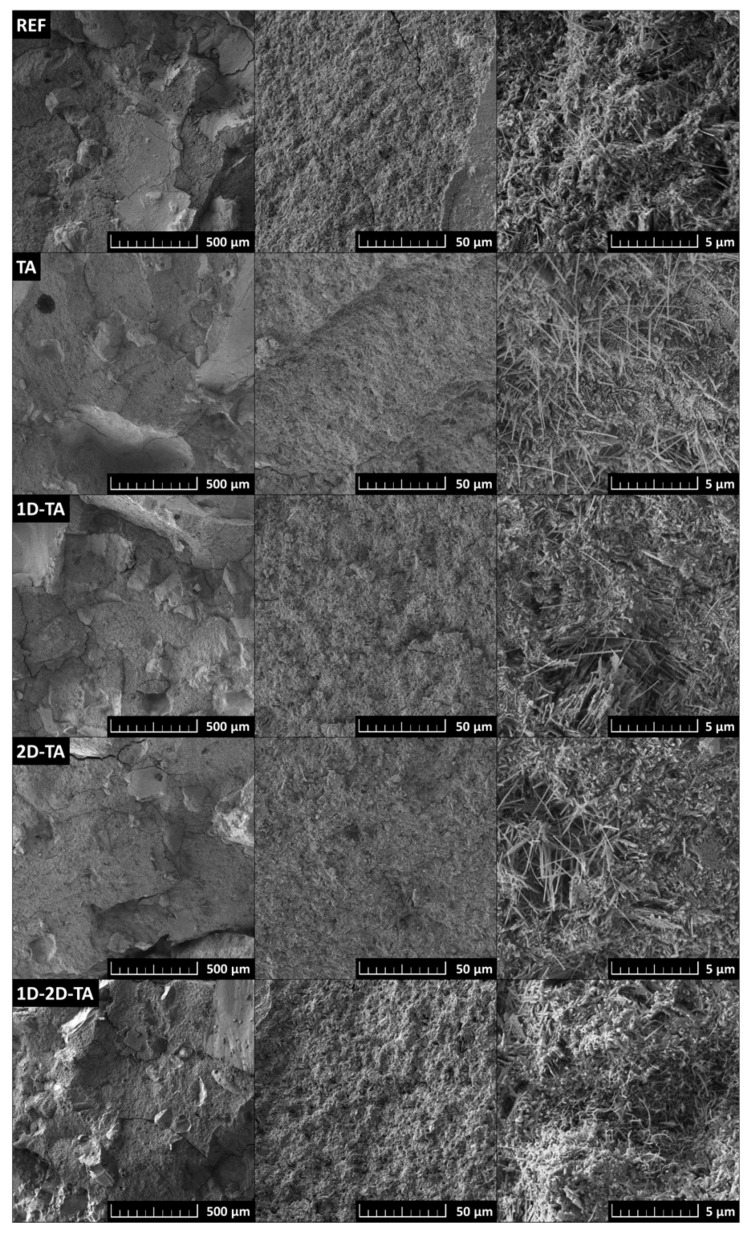
SEM micrographs of prepared MOC-based composites.

**Figure 8 polymers-15-04300-f008:**
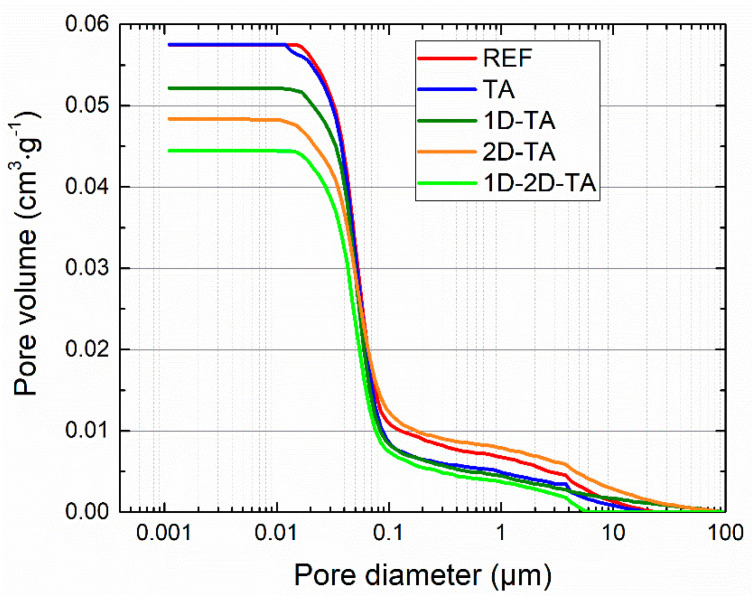
Cumulative pore size distribution of the prepared MOC-based composites.

**Figure 9 polymers-15-04300-f009:**
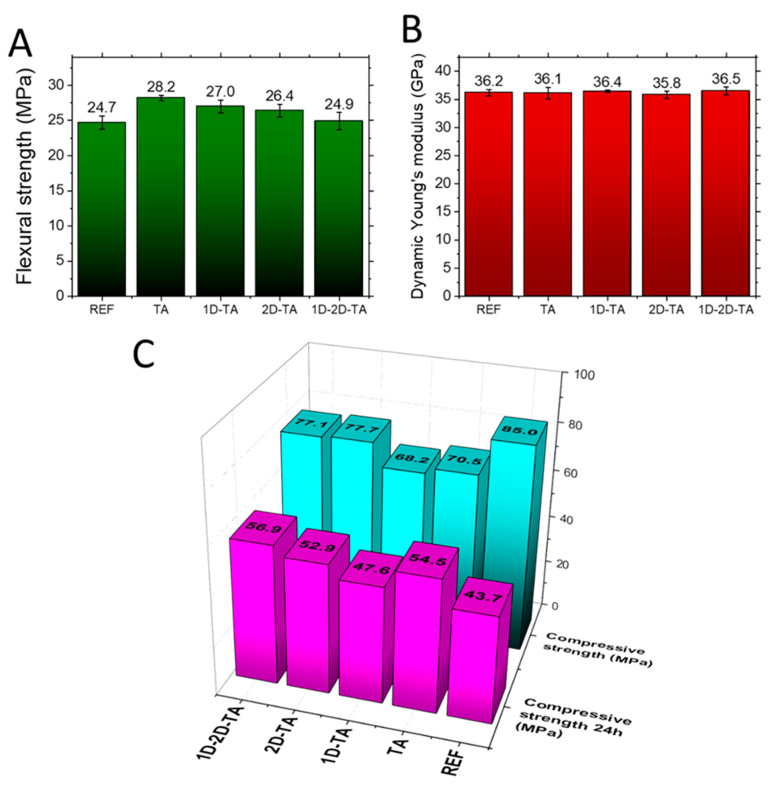
Mechanical parameters of prepared samples: (**A**) flexural strength, (**B**) Dynamic Young’s modulus, and (**C**) comparison of compressive strength before and after 24 h water immersion.

**Figure 10 polymers-15-04300-f010:**
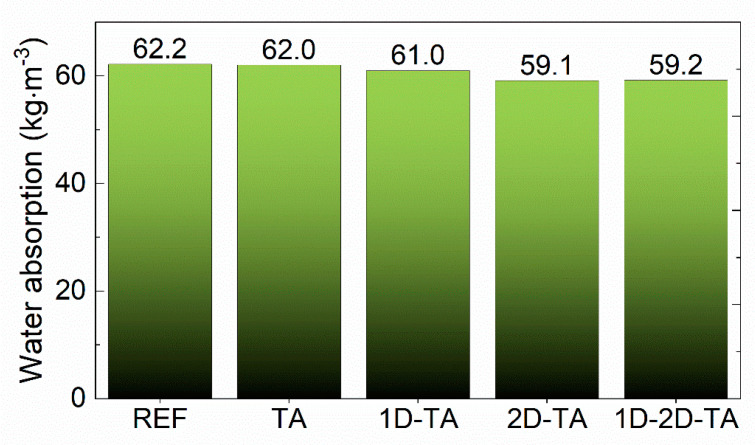
Water absorption of prepared samples after 24 h immersion.

**Table 1 polymers-15-04300-t001:** Fresh mixture properties.

Mixture	Mass (g)								
	MgO	MgCl_2_∙6H_2_O	Water	PG1	PG2	PG3	TA	1D	2D
PASTE	450.0	453.97	281.60	-	-	-	-	-	-
REF	450.0	453.97	281.60	450.0	450.0	450.0	-	-	-
TA	450.0	453.97	281.60	450.0	450.0	450.0	2.37	-	-
1D-TA	450.0	453.97	281.60	450.0	450.0	450.0	2.37	5.92	-
2D-TA	450.0	453.97	281.60	450.0	450.0	450.0	2.37	-	5.92
1D-2D-TA	450.0	453.97	281.60	450.0	450.0	450.0	2.37	2.96	2.96

**Table 2 polymers-15-04300-t002:** Chemical composition of prepared MOC-based composites obtained from EDS.

Mixture	wt. %				
	O	Mg	Si	Cl	C
REF	41.5	21.6	13.1	12.4	11.4
TA	39.4	24.3	11.2	13.8	11.2
1D-TA	42.6	18.6	17.5	10.2	11.1
2D-TA	38.5	26.3	6.8	16.1	12.3
1D-2D-TA	39.8	26.1	6.6	14.3	13.2

**Table 3 polymers-15-04300-t003:** Structural parameters of prepared MOC-based composites.

Mixture	Bulk Density	Specific Density	Porosity (MIP)	Total Pore Volume	Average Pore Diameter
	(kg·m^−3^)	(kg·m^−3^)	(%)	(cm^−3^·g^−1^)	(μm)
REF	2029 ± 28	2251 ± 27	10.6 ± 0.2	0.052	0.047
TA	2030 ± 28	2260 ± 27	10.5 ± 0.2	0.058	0.043
1D-TA	2035 ± 28	2269 ± 27	8.8 ± 0.2	0.045	0.046
2D-TA	2031 ± 28	2279 ± 27	10.5 ± 0.2	0.058	0.049
1D-2D-TA	2035 ± 29	2266 ± 27	9.2 ± 0.2	0.048	0.049

## Data Availability

The data presented in this study are available on request from the corresponding author.

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
