# Peer review of "MOC Composites for Construction: Improvement in Water Resistance by Addition of Nanodopants and Polyphenol"

_polymers, 2023, doi:10.3390/polym15214300_

Round 1

Reviewer 1 Report

Comments and Suggestions for Authors

The manuscript is well written. Below are few minor possible modificatins:

State the reason for the achievement of higher flexural strength. A detailed explanation should be provided.

Images depicting the flexural strength test, compressive strength test, and water absorption test should be included.

Error bars should be included in the presentation of compressive strength test results.

Enhancements in durability and mechanical characteristics achieved through the use of various additives should be expressed in terms of a percentage in the Conclusions.

Comments on the Quality of English Language

Please go through the grammar across the text.

Author Response

Dear reviewer, thank you very much for your positive evaluation of our manuscript. It is very much appreciated. We have addressed all your comments and suggestions in the revised manuscript. We believe that the corrections we made have improved the overall presentation of our research. The changes we made are highlighted in green. See the specific comments below.

The manuscript is well written. Below are few minor possible modifications:

State the reason for the achievement of higher flexural strength. A detailed explanation should be provided.

The increase in the flexural strength is the result of three effects:  i) high intrinsic tensile strength of the used nanoadditives themselves, ii) bridging effect of 2D nanodopant, iii) decrease in the volume of pores having diameter > 0.1 μm. The last effect was not identified for 2D-TA sample, but was well visible for materials TA, 1D-TA and 1D-2D-TA.

The text was revised accordingly and new reference was added.

Images depicting the flexural strength test, compressive strength test, and water absorption test should be included.

As recommended, images describing the performed tests of the flexural strength, compressive strength, and water absorption are newly added in the revised manuscript. 

Enhancements in durability and mechanical characteristics achieved through the use of various additives should be expressed in terms of a percentage in the Conclusions.

The text was revised and the information on the improvement of durability parameter (softening coefficient) and flexural strength in terms of a percentage was completed in the section Conclusions. 

Please go through the grammar across the text.

The language and grammar were thoroughly revised throughout the manuscript.

Reviewer 2 Report

Comments and Suggestions for Authors

I have reviewed the revised manuscript and acknowledged that the authors had made substantial revisions. However, there are still some minor questions in this manuscript. Therefore, I would say that the paper should be accepted for publication after answering the following questions

1.       The quality of English needs improving.

2.       The innovation of the article is not clear enough. Could the authors point out the innovation of this paper compared with the previous research?

3.       In this paper, there is no test equipment diagram and schematic diagram of the experiment process. Please add the experiment equipment diagram and schematic diagram and elaborate on them.

4.       The introduction needs to be updated and rewritten. Authors have to make a good discussion from the nearest research and mention the novelty of their work.

5.       Why use tannic acid as an additive instead of other materials?

6.       What are the advantages or properties of tannic acid?

7.       What's the pattern of the elements map of EDS? What does it say?

8.       Why do the needles of tannic acid samples become thinner?

9.       Why do you think tannic acid can distribute nanoparticles uniformly in the MOC structure?

10.   In Figure 8, why does the compressive strength of REF become much lower after 24 h water immersion?

11.   There is no index to describe the extent to which tannic acid improves the water resistance of MOC composites, which should be expressed in the conclusion.

Author Response

I have reviewed the revised manuscript and acknowledged that the authors had made substantial revisions. However, there are still some minor questions in this manuscript. Therefore, I would say that the paper should be accepted for publication after answering the following questions

Dear reviewer 2, thank you for your valuable comments and remarks. We implemented them in the revised manuscript as advised. We believe that the corrections we made have improved the overall presentation of our research. The changes we made are highlighted in green. See the specific comments below.

1. The quality of English needs improving.

The language and grammar were thoroughly revised throughout the manuscript.

2. The innovation of the article is not clear enough. Could the authors point out the innovation of this paper compared with the previous research?

The novelty of our approach and the innovation was pointed out more clearly in the revised manuscript.

3. In this paper, there is no test equipment diagram and schematic diagram of the experiment process. Please add the experiment equipment diagram and schematic diagram and elaborate on them.

The experiment diagram was added to the Materials and methods section.

4. The introduction needs to be updated and rewritten. Authors have to make a good discussion from the nearest research and mention the novelty of their work.

The introduction section was updated and extended accordingly. New references to recent research in this field were added.

5. Why use tannic acid as an additive instead of other materials?

The reason for the use of tannic acid was clarified in the updated Introduction section.

6. What are the advantages or properties of tannic acid?

The properties and advantages of tannic acid were described in the updated Introduction section.

7. What's the pattern of the elements map of EDS? What does it say?

The explanation of the sample patterns shown in the EDS maps was added to the Results section of the revised manuscript.

8. Why do the needles of tannic acid samples become thinner?

The thinning of the MOC needles was explained in the results sections.

9. Why do you think tannic acid can distribute nanoparticles uniformly in the MOC structure?

The uniform distribution of the nanoparticles in the MOC structure due to the addition of tannic acid is caused by its adsorption onto the carbon-based nanomaterial creating a π-π interaction between the aromatic rings of tannic acid and carbon atoms in the structure of the nanomaterial. This increases the steric repulsion between the individual carbon nanomaterial particles, which makes them easier to disperse. This was clarified in the Introduction section.

10. In Figure 8, why does the compressive strength of REF become much lower after 24 h water immersion?

The dramatic drop in the compressive strength of the reference material is the result of the decomposition of MOC Phase 5 into Mg(OH)2, which has much lower mechanical strength. Within the water induced structural changes, the mechanical interlocking of a stable need-like crystallized Phase 5 is damaged, which is the reason of the loss of mechanical integrity. This mechanism is generally attributed with the pure water resistance of MOC-based products. The text was revised and new references were added.

11. There is no index to describe the extent to which tannic acid improves the water resistance of MOC composites, which should be expressed in the conclusion.

This information was added to the conclusion section of the revised manuscript.

Round 2

Reviewer 1 Report

Comments and Suggestions for Authors

The suggested modifications are appearing in the manuscript.